# Towards Unsupervised Dense Information Retrieval with Contrastive Learning

## Abstract

Information retrieval is an important component in natural language processing, for knowledge intensive tasks such as question answering and fact checking. Recently, information retrieval has seen the emergence of dense retrievers, based on neural networks, as an alternative to classical sparse methods based on term-frequency. These models have obtained state-of-the-art results on datasets and tasks where large training sets are available. However, they do not transfer well to new domains or applications with no training data, and are often outperformed by term-frequency methods such as BM25 which are not supervised. Thus, a natural question is whether it is possible to train dense retrievers without supervision. In this work, we explore the limits of contrastive learning as a way to train unsupervised dense retrievers, and show that it leads to strong retrieval performance. More precisely, we show on the BEIR benchmark that our model outperforms BM25 on 8 out of 14 datasets. Furthermore, when a few thousands examples are available, we show that fine-tuning our model on these leads to strong improvements compared to BM25. Finally, when used as pre-training before fine-tuning on the MS MARCO dataset, our technique obtains state-of-the-art results on the BEIR benchmark.

## 1 Introduction

Document retrieval is the task of finding relevant documents in a large collection to answer specific queries. This is an important task by itself and a core component of many natural language processing (NLP) problems, such as open domain question answering (Chen et al., 2017) or fact checking (Thorne et al., 2018). Traditionally, retrieval systems, or retrievers, leverage lexical similarities to match queries and documents, using, for instance, TF-IDF or BM25 weighting (Robertson & Zaragoza, 2009). These approaches, based on exact matches between tokens of the queries and documents, suffer from the lexical gap and do not generalize well (Berger et al., 2000). By contrast, approaches based on neural networks allow learning beyond lexical similarities, resulting in state-of-the-art performance on question answering benchmarks, such as MS-MARCO (Nguyen et al., 2016) or NaturalQuestion (Kwiatkowski et al., 2019).

As training neural networks requires large amount of data, their strong retrieval results were possible for domains and applications where large training datasets are available. In the case of retrieval, creating these datasets requires manually matching queries to the relevant documents in the collection. This is hardly possible when the collection contains millions or billions of element, resulting in many scenarios where only a few in-domain examples, if any, are available. A potential solution is to train a dense retriever on a large retrieval dataset such as MS MARCO, and then apply it to new domains, a setting refered to as *zero-shot*. Unfortunately, in this setting dense retrievers are often outperformed by classical methods based on term-frequency, which do not require supervision (Thakur et al., 2021).

Thus, a natural alternative to transfer learning is unsupervised learning, which raises the following question: *is it possible to train dense retrievers without supervision, and match the performance of BM25?* Training dense retrievers without supervision can be achieved by using a pretext task that approximates retrieval. Given a document, one can generate a synthetic query and then train the network to retrieve the original document, among many others, given the query. The inverse Cloze task (ICT), proposed by Lee et al. (2019) to pre-train retrievers, uses a given sentence as query and predict the context surrounding it. While showing promising results as pre-training (Chang et al., 2020; Sachan et al., 2021a), this approach still lags behind BM25 when used as a zero-shot retriever.

ICT is strongly related to contrastive learning (Wu et al., 2018), which has been widely applied in computer vision (Chen et al., 2020; He et al., 2020). In particular, computer vision models trained with the latest contrastive learning methods lead to features well suited to retrieval (Caron et al., 2021). In this work, we thus propose to revisit how well contrastive learning performs to train dense retrievers without supervision. We want to investigate how much the recent developments from computer vision, such as MoCo, can improve dense retrievers. Here, our goal is not to develop new techniques, but to determine how far we can go by pushing existing methods to their limits.

In this paper, we thus make the following contributions. First, we show that contrastive learning can lead to strong unsupervised retrievers: our model achieves Recall@100 results competitive with BM25 on most of the BEIR benchmark. Second, in a few-shot setting, we show that our model benefits from few training examples, and obtains better results than transfering models from large datasets such as MS MARCO. Third, when used as a pre-training method before fine-tuning on MS MARCO, our technique leads to state-of-the-art retrieval performance on the BEIR benchmark. Finally, we perform ablations to motivate our design choices, and show that cropping works better than the inverse Cloze task.

## 2 RELATED WORK

In this section, we briefly review relevant work in information retrieval, and application of machine learning to this problem. This is not an exhaustive review, and we refer the reader to Manning et al. (2008), Mitra et al. (2018) and Lin et al. (2020) for a more complete introduction to the field.

**Term-frequency based information retrieval.** Historically, in information retrieval, documents and queries are represented as sparse vectors where each element of the vectors corresponds to a term of the vocabulary. Different weighing schemes have been proposed, to determine how important a particular term is to a document in a large dataset. One of the most used weighing scheme is known as TF-IDF, and is based on inverse document frequency, or term specificity (Jones, 1972). BM25, which is still widely used today, extends TF-IDF (Robertson et al., 1995). A well known limitation of these approaches is that they rely on exact match to retrieve documents. This lead to the introduction of latent semantic analysis (Deerwester et al., 1990), in which documents are represented as low dimensional dense vectors.

**Neural network based information retrieval.** Following the successful application of deep learning methods to natural language processing, neural networks techniques were introduced for information retrieval. Huang et al. (2013) proposed a deep bag-of-words model, in which representations of queries and documents are computed independently. A relevance score is then obtained by taking the dot product between representations, and the model is trained end-to-end on click data from a search engine. This method was later refined by replacing the bag-of-words model by convolutional neural networks (Shen et al., 2014) or recurrent neural network (Palangi et al., 2016). A limitation of bi-encoders is that queries and documents are represented by a single vector, preventing the model to capture fine-grained interactions between terms. Nogueira & Cho (2019) thus introduced a cross-encoder model, based on a pre-trained BERT model (Devlin et al., 2019), which jointly encodes queries and documents. The application of a strong pre-trained model, as well as the cross-encoder architecture, lead to important improvement on the MS-MARCO benchmark (Bajaj et al., 2016).

The methods described in the previous paragraph were applied to re-rank documents, which were retrieved with a traditional IR system such as BM25. Gillick et al. (2018) first studied whether continuous retrievers, based on bi-encoder neural models, could be viable alternative to re-ranking. In the context of question answering, Karpukhin et al. (2020) introduced a dense passage retriever (DPR) based on the bi-encoder architecture. This model is initialized with a BERT network, and trained discriminatively using pairs of queries and relevant documents, with hard negatives from BM25. Xiong et al. (2020) further extended this work by mining hard negatives with the model itself during optimization, and trained on the MS-MARCO dataset. Once a collection of documents, such as Wikipedia articles, is encoded, retrieval is performed with a fast k-nearest neighbors library such as FAISS Johnson et al. (2019). To alleviate the limitations of bi-encoders, Humeau et al. (2019) introduces the poly-encoder architecture, where documents are encoded by multiple vectors. Similarly, Khattab et al. (2020) proposes the ColBERT model, which keeps a vector representation for each term of the queries and documents. To make the retrieval tractable, the term-level function

is approximated to first retrieve an initial set of candidates, which are then re-ranked with the true score. In the context of question answering, knowledge distillation has been used to train retrievers, either using the attention scores of the reader of the downstream task as synthetic labels (Izacard & Grave, 2021), or the relevance score from a cross encoder (Yang & Seo, 2020). Luan et al. (2020) compares, theoretically and empirically, the performance of sparse and dense retrievers, including bi-, cross- and poly-encoders. Dense retrievers, such as DPR, can lead to indices that weigh close to 100 GB when encoding document collections such as Wikipedia. Izacard et al. (2020) shows how to compress such indices, with limited impact on performance, making them more practical to use.

**Self-supervised learning for NLP.** Following the success of word2vec (Mikolov et al., 2013), many self-supervised techniques have been proposed to learn representation of text. Here, we briefly review the ones that are most related to our approach: sentence level models and contrastive techniques. Jernite et al. (2017) introduced different objective function to learn sentence representations, including next sentence prediction and binary order prediction. These objectives were later used in pre-trained models based on transformers, such as BERT (Devlin et al., 2019) and AlBERT (Lan et al., 2019). In the context of retrieval, Lee et al. (2019) introduced the inverse cloze task (ICT), whose purpose is to predict the context surrounding a span of text. Guu et al. (2020) integrated a bi-encoder retriever model in a BERT pre-training scheme. The retrieved documents are used as additional context in the BERT task, and the whole system is trained end-to-end in an unsupervised way. Similarly, Lewis et al. (2020) proposed to jointly learn a retriever and a generative seq2seq model, using self-supervised training. Chang et al. (2020) compares different pre-training tasks for retrieval, including the inverse cloze task. Constrastive learning was introduced in computer vision by Wu et al. (2018), followed by several modifications to improve the training (He et al., 2020; Chen et al., 2020; Caron et al., 2020). In the context of natural language processing, Fang et al. (2020) proposed to apply MoCo where positive pairs of sentences are obtained using back-translation. Different works augmented the masked language modeling objective with a contrastive loss (Giorgi et al., 2020; Wu et al., 2020; Meng et al., 2021). Finally, SBERT (Reimers & Gurevych, 2019) uses a siamese network similar to contrastive learning to learn a BERT-like model that is adapted to matching sentence embeddings. Their formulation is similar to our work but requires aligned pairs of sentences to form positive pairs while we propose to use data augmentation to leverage large unaligned textual corpora. Concurrent to this work, Gao & Callan (2021) have also shown the potential of contrastive learning for information retrieval; building on the same observation that both tasks share a similar structure.

## 3   METHOD

In this section, we describe how to train a dense retriever with no supervision. We review the model architecture and then describe contrastive learning - a key component of its training.

The objective of a retriever is to find relevant documents in a large collection for a given query. Thus, the retriever takes as input the set of documents and the query and outputs a relevance score for each document. A standard approach is to encode each query–document pair with a neural network (Nogueira & Cho, 2019). This procedure requires re-encoding every document for any new query and hence does not scale to large collections of documents. Instead, we propose to use a bi-encoder architecture, where documents and queries are encoded independently (Huang et al., 2013; Karpukhin et al., 2020). One can compute the relevance score by taking the dot product (or cosine similarity) between the document's representation and the representation of the query. More precisely, given a pair of query $q$ and document $d$, we encode each of them independently using the same model, $f_\theta$, parametrized by $\theta$. The relevance score $s(q, d)$ is then the dot product of the resulting representations:

$$s(q, d) = \langle f_\theta(q), f_\theta(d) \rangle.$$

In practice, we use a transformer network for $f_\theta$ to embed both queries and documents. The representation $f_\theta(q)$ (resp. $f_\theta(d)$) for a query (resp. document) is obtained by averaging the hidden representations of the last layer. Following previous work on dense retrieval with neural networks, we use the BERT base uncased architecture and refer the reader to Devlin et al. (2019) for more details.

### 3.1 UNSUPERVISED TRAINING ON UNALIGNED DOCUMENTS

In this section, we describe our unsupervised training pipeline. We briefly review the loss function traditionally used in contrastive learning. We then discuss obtaining positive pairs from a single text document, a critical ingredient for this training paradigm.

#### 3.1.1 CONTRASTIVE LEARNING

Contrastive learning is an approach that relies on the fact that every document is, in some way, unique. This signal is the only information available in the absence of manual supervision. The resulting algorithm learns by discriminating between documents, using a contrastive loss (Wu et al., 2018). This loss compares either positive (from the same document) or negative (from different documents) pairs of document representations. More formally, given a positive pair of representations $(q, k_+)$ and a set of negative pairs $(q, k_i)_{i=0..K}$, the contrastive InfoNCE loss is defined as:

$$\mathcal{L}(q, k_+) = \frac{\exp(s(q, k_+)/\tau)}{\sum_{i=0}^{K} \exp(s(q, k_i)/\tau)},$$

where $\tau$ is a temperature parameter. This loss encourages the relevance score of similar examples to be high and that of dissimilar examples to be low. Another interpretation of this loss function is the following: given the query representation $q$, the goal is to recover, or retrieve, the representation $k_+$ corresponding to the positive document, among all the negatives $k_i$. In the following, we refer to the left-hand side representations in the score $s$ as queries and the right-hand side representations as keys.

#### 3.1.2 BUILDING POSITIVE PAIRS FROM A SINGLE DOCUMENT

A crucial element of contrastive learning is how to build positive pairs from a single input. In computer vision, this step relies on applying two independent data augmentations to the same image, resulting in two "views" that form a positive pair (Wu et al., 2018; Chen et al., 2020). While we consider similar independent text transformation, we also explore dependent transformations designed to reduce the correlation between views.

**Inverse Cloze Task** is a data augmentation that generates two mutually exclusive views of a document, introduced in the context of retrieval by Lee et al. (2019). The idea is to take a span of tokens to form one view, and its complementary to form the other view. More precisely, given a sequence of text $(w_1, ..., w_n)$, ICT samples a span $a, b$, where $1 \leq a < b \leq n$, uses the tokens of the span as the query and the rest of the tokens as the document (or key). In the original implementation by Lee et al. (2019) the span corresponds to a sentence, and is kept in the document 10% of the time to encourage lexical matching.

**Independent cropping** is a common independent data augmentation used for images where views are generated independently by cropping the input. In the context of text, cropping is equivalent to sampling a span of tokens. This strategy thus samples independently two spans from a document to form a positive pair. As opposed to the inverse Cloze task, in *cropping* both views of the example correspond to contiguous subsequence of the original data. A second difference between cropping and ICT is the fact that the task is symmetric: both the queries and documents follow the same distribution. Independent cropping can also lead to overlap between the two views of the data, hence encouraging the network to learn exact matches between the query and document, in a way that is similar to lexical matching methods like BM25. In practice, we can either fix the length of the span for the query and the key, or sample them.

**Additional data augmentation.** Finally, we also consider additional data augmentations such as random word deletion, random word replacement or random word masking. We use these perturbation in addition to random cropping.

#### 3.1.3 BUILDING LARGE SET OF NEGATIVE PAIRS

An important aspect of contrastive learning is to maintain a large set of negative pairs. Most standard frameworks differ from how the negatives are handled, and we briefly describe two of them, SimCLR and MoCo, that we use in this work.

**Negative pairs within a batch.** A first solution, called SimCLR (Chen et al., 2020), is to generate the negative pairs by using the other examples from the same batch: each example in a batch is transformed twice to generate positive pairs, and we generate negative pairs by using the views from the other examples in the batch. In that case, the gradient is back-propagated through the representations of both the queries and the keys. A downside of this approach is that it requires extremely large batch sizes to work well. This technique was also considered when using ICT to pre-train retrievers by Lee et al. (2019).

**Negative pairs across batches.** An alternative approach is to store representations from previous batches in a queue and use them as negative examples in the loss (Wu et al., 2018). This allows for smaller batch size but slightly changes the loss by making it asymmetric between "queries" (one of the view generated from the elements of the current batch), and "keys" (the other generated views, as well as the elements stored in the queue). Gradient is only backpropagated through the "queries", and the representation of the "keys" are considered as fixed. In practice, the features stored in the queue from previous batches comes form previous iterations of the network. This leads to a drop of performance when the network rapidly changes during training. Instead, He et al. (2020) proposed to generate representations of keys from a second network that is updated more slowly. This approach, called MoCo, considers two networks: one for the keys, parametrized by $\theta_k$, and one of the query, parametrized by $\theta_q$. The parameters of the query network are updated with backpropagation and stochastic gradient descent, like in SimCLR, while the parameters of the key network, or Momentum encoder, is updated from the parameters of the query network by using a exponential moving average:

$$\theta_k \leftarrow m\theta_k + (1-m)\theta_q,$$

where $m$ is the momentum parameter that takes its value in $[0,1]$.

## 4 EXPERIMENTS

In this section, we empirically evaluate contrastive learning as a way to train dense retrievers. First, we compare our best models to the state-of-the-art on competitive retrieval benchmarks for both zero-shot and few-shot retrieval. Then, we provide an ablation study to motivate the technical choices leading to our best retriever, shedding some light on the important ingredients of our approach. We give technical details about our models in Appendix A.

### 4.1 BEIR BENCHMARK

The BEIR benchmark, introduced by Thakur et al. (2021), contains 18 retrieval datasets with a focus on diversity. Each dataset is made of a set of queries, the corresponding relevant documents and a large collection of documents to retrieve from. These datasets correspond to nine different retrieval tasks, such as fact checking, question answering, entity retrieval or citation prediction. Moreover, the benchmark covers a large number of domains, such as Wikipedia, news articles, scientific publications or social media posts. Finally, there is also diversity in terms of documents and queries length, with queries ranging from 3 to 190 words and documents from 11 to 630 words. Most datasets from the BEIR benchmark do not contain a training set, and the focus of the benchmark is *zero-shot retrieval*. However, most machine learning based retrievers are still trained on supervised data, such as the large scale retrieval dataset MS-MARCO (Bajaj et al., 2016). Following standard practice, we report two metrics on this benchmark, the nDGC@10 and the Recall@100. These two metrics are complementary and both important: nDCG focuses on the ranking of the top 10 retrieved documents, and is good at evaluating documents returned to human, for example in a search engine. On the other hand, the Recall@100 is relevant to evaluate retrievers that are included in marchine learning systems, such as question answering. Indeed, such models can process hundreds of documents, and ignore the ranking of these documents (Izacard & Grave, 2020).

### 4.2 BASELINES

First, we compare our approach to the BM25 ranking function, which does not require any training data. However, the relevance of individual terms depends on the inverse document frequency, which is computed over each document collection independently. In that sense, the ranking function used in the BM25 baseline is adapted to each dataset on which retrieval is performed. We also compare

| | BM25 | ColBERT | SPARTA | docT5Query | DPR | ANCE | GenQ | TAS-B | Ours |
|---|---|---|---|---|---|---|---|---|---|
| TREC-COVID | 65.6 | 67.7 | 53.8 | **71.3** | 33.2 | 65.4 | 61.9 | 48.1 | 61.0 |
| NFCorpus | 32.5 | 30.5 | 30.1 | 32.8 | 18.9 | 23.7 | 31.9 | 31.9 | **33.2** |
| NQ | 32.9 | **52.4** | 39.8 | 39.9 | 47.4 | 44.6 | 35.8 | 46.3 | 50.2 |
| HotpotQA | 60.3 | 59.3 | 49.2 | 58.0 | 39.1 | 45.6 | 53.4 | 58.4 | **64.5** |
| FiQA | 23.6 | **31.7** | 19.8 | 29.1 | 11.2 | 29.5 | 30.8 | 30.0 | 28.8 |
| ArguAna | 31.5 | 23.3 | 27.9 | 34.9 | 17.5 | 41.5 | **49.3** | 42.9 | 46.0 |
| Tóuche-2020 | **36.7** | 20.2 | 17.5 | 34.7 | 13.1 | 24.0 | 18.2 | 16.2 | 25.9 |
| Quora | 78.9 | **85.4** | 63.0 | 80.2 | 24.8 | 85.2 | 83.0 | 83.5 | **85.4** |
| CQADupStack | 29.9 | **35.0** | 25.7 | 32.5 | 15.3 | 29.6 | 34.7 | 31.4 | 32.2 |
| DBPedia | 31.3 | **39.2** | 31.4 | 33.1 | 26.3 | 28.1 | 32.8 | 38.4 | 38.8 |
| Scidocs | 15.8 | 14.5 | 12.6 | **16.2** | 7.7 | 12.2 | 14.3 | 14.9 | 16.0 |
| Fever | 75.3 | 77.1 | 59.6 | 71.4 | 56.2 | 66.9 | 66.9 | 70.0 | **77.7** |
| Climate-FEVER | 21.3 | 18.4 | 8.2 | 20.1 | 14.8 | 19.8 | 17.5 | 22.8 | **23.4** |
| SciFact | 66.5 | 67.1 | 58.2 | 67.5 | 31.8 | 50.7 | 64.4 | 64.3 | **68.1** |
| Average | 43.0 | 44.4 | 35.5 | 44.4 | 25.5 | 40.5 | 42.5 | 42.8 | **46.5** |
| Best on | 1 | 5 | 0 | 2 | 0 | 0 | 1 | 0 | **6** |

Table 1: **BEIR Benchmark.** We report nDCG@10 on the test sets of the datasets from the BEIR benchmark for bi-encoder methods without re-ranker. We also report the average ("Average") and number of datasets where a method is the best ("Best on") over the entire BEIR benchmark (excluding three datasets because of their licence). We evaluate retrievers after pre-training on unsupervised data with our contrastive learning and finetuning on MS-MARCO ("Ours"). Bold is the best overall.

to machine learning based retrievers, which can be classified in three categories: sparse, dense and late-interaction. For sparse methods, we compare to *docT5query* (Nogueira et al., 2019), which is a document expansion technique based on the T5 model (Raffel et al., 2019). The second sparse model we use as baseline is *SPARTA* (Zhao et al., 2020), which computes sparse representations of documents using a BERT model. More precisely, the ranking function of SPARTA is based on interactions between terms of the document and the query. Because the representations of the query terms are non-contextualized, the scores can be pre-computed when indexing the database. For dense methods, we use *DPR* (Karpukhin et al., 2020) and *ANCE* (Xiong et al., 2020) as baselines, which are bi-encoder models trained on supervised data such as NaturalQuestions or MS-MARCO. We also include the *TAS-B* technique (Hofstätter et al., 2021), which performs distillation from a cross-encoder to a bi-encoder model. Finally, we compare to GenQ, which uses a generative query model to create synthetic query-document pairs, and is applied to each dataset. Thus, a different GenQ model is used for each dataset of the BEIR benchmark. For late-interaction methods, we use ColBERT (Khattab et al., 2020), which computes pairwise scores between contextualized representations of the query and the document. To make this approach scalable, this scoring function is only applied on documents obtained by applying an approximation of the full score. Interestingly, it should be noted that BM25 is a very competitive baseline in the zero-shot setting, outperforming all dense retrievers on average over the BEIR benchmark. Only the re-ranking models and the query expansion (docT5query), which is also based on BM25, obtain better results than BM25.

### 4.3 RESULTS

In Table 1, we report the nDCG@10 on the BEIR benchmark for different retrievers. The corresponding Recall@100 performances can be found in Table 7 of appendix. We individually report results on each dataset as well as the average ("Avg.") over 14 datasets of the BEIR Benchmark (excluding 3 for license reasons). We omit retrievers that use a re-ranker during inference since it is orthogonal to our focus and can be applied to any method.

We observe that contrastive pre-training leads to strong performance among retrievers: our model obtains state-of-the art performance when finetuned on the MS-MARCO dataset. It should be noted that our finetuning procedure on MS-MARCO is simpler than for other dense retrievers: we use a simple strategy for negative mining and do not use distillation. Our model would probably also benefits from improvements proposed by these retrievers, but this is beyond the scope of this paper.

|  | finetuning | SciFact | NFCorpus | FiQA |
|---|---|---|---|---|
| Dataset size |  | 809 | 2,590 | 5,500 |
| BM25 | - | 66.5 | 32.5 | 23.6 |
| BERT | - | 75.2 | 29.9 | 26.1 |
| Ours | - | 84.0 | 33.9 | 31.5 |
| BERT | MS-MARCO | 80.9 | 33.2 | 30.9 |
| Ours | MS-MARCO | **84.1** | **34.9** | **34.4** |

Table 2: **Few-shot retrieval.** nDCG@10 on the test sets of benchmarks when finetuned on a small training set. We compare BERT and our pre-training with and without the finetuning step on MS-MARCO. Note that our unsupervised pre-training alone leads to better performance than BERT finetuned on MS-MARCO.

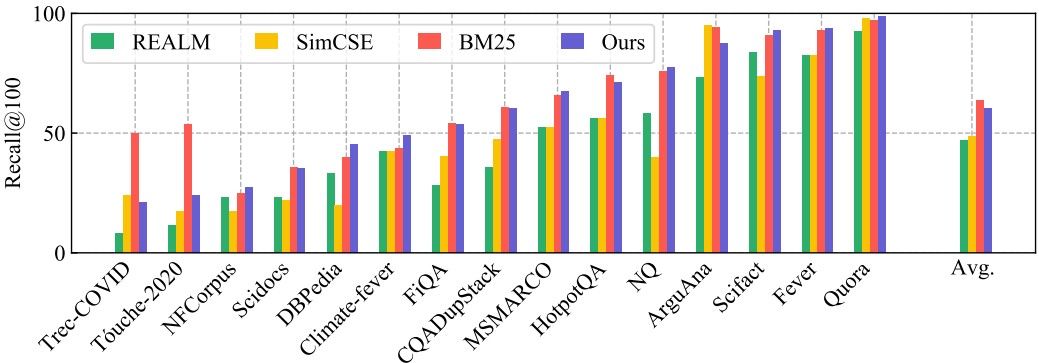

Figure 1: **Zero-shot retrieval.** We compare our pre-training without using *any* annotated data to REALM, SimCSE and BM25. For SimCSE we report results of the model using RoBERTa large. REALM uses annotated entity recognition data for training. We highlight that our unsupervised pretraining is on par with BM25 but on 2 datasets.

We also illustrate the benefit of our dense retriever compared to BM25 by evaluating it in a *few shot* setting, where we have access to a small number of training queries and relevant documents to finetune locally the retriever. This setting is common in practice, and lexical based methods, like BM25, cannot benefit leverage the small training sets to adapt its weights. In Table 2, we report the nDCG@10 results on the three datasets from the BEIR benchmark associated with the smallest training sets. These training sets range from 809 to almost 5,500 queries. We observe that on these small datasets, our pre-training leads to significantly better results than BERT pre-training only, even when BERT model is finetuned on MS-MARCO. Our pre-trained model also outperforms BM25, showing the advantage of dense retriever over lexical methods in the low-shot setting.

Finally, we compare the performance of fully unsupervised models, i.e., without finetuning on MS-MARCO or other annotated data. In Table 8 we report the retrieval performance of our model on two standard question answering benchmarks: NaturalQuestions (Kwiatkowski et al., 2019) and TriviaQA (Joshi et al., 2017). In Figure 1 we report the performance of unsupervised models on the BEIR benchmark. Interestingly, we observe that, in this setting, our pre-training can lead to a model with competitive retrieval performance compared to BM25. Our approach outperforms previously proposed unsupervised dense retrievers. On question answering benchmarks it also outperforms a strong BM25 baseline (Ma et al., 2021) by 3% for the Recall@100, and is on par with BM25 on TriviaQA for the Recall@100. On datasets of the BEIRbenchmark, we observe that our unsupervised retriever is competitive with BM25 on every datasets, but TREC-COVID and Tóuche-2020. These results show the potential of contrastive learning to develop fully unsupervised dense retrievers.

## 5 ABLATION STUDIES

In this section, we investigate the influence of different design choices on our method. In these ablations, all the models are pre-trained on Wikipedia for 200k gradient steps, with a batch size of

2,048 (on 32 GPUs). Each fine-tuning on MS-MARCO takes 20k gradient steps with a batch size of 512 (on 8 GPUs), using AdamW and no hard negatives.

**MoCo vs. SimCLR.** First, we compare the two contrastive pre-training methods: MoCo and SimCLR. As in SimCLR, the number of negative examples is equal to the batch size, we train models with a batch size of 4,096 and restrict the queue in MoCo to the same number of elements. This experiment measures the effect of using of momentum encoder for the keys instead of the same network as for the queries. Using a momentum also prevents from backpropagating the gradient through the keys. We report results, with fine-tuning on MS-MARCO in Table 5. We observe that the difference of performance between the two methods is small, especially after fine-tuning on MS-MARCO. This validates our choice of MoCo as our contrastive learning framework, since it scales to a larger number of negative examples without the need to increase the batch size.

|        | NFCorpus | NQ   | FiQA | ArguAna | Quora | DBPedia | SciDocs | FEVER | AVG  |
|--------|----------|------|------|---------|-------|---------|---------|-------|------|
| MoCo   | 31.4     | 43.8 | 26.3 | 38.4    | 83.5  | 40.1    | 16.1    | 73.6  | 42.8 |
| SimCLR | 29.5     | 39.5 | 23.6 | 38.3    | 83.7  | 36.1    | 16.1    | 73.4  | 41.6 |

Table 3: **MoCo vs. SimCLR.** In this table, we report nDCG@10 on the BEIR benchmark for SimCLR and MoCo, after fine-tuning on the MS-MARCO dataset.

**Number of negative examples.** Next, we study the influence of the number of negatives used in the contrastive loss, by varying the queue size of the MoCo algorithm. We consider values ranging from 2,048 to 131,072, and report results in Figure 2. We see that on average over the BEIR benchmark, increasing the number of negatives leads to better retrieval performance, especially in the unsupervised setting. However, we note that this effect is not equally strong for all datasets.

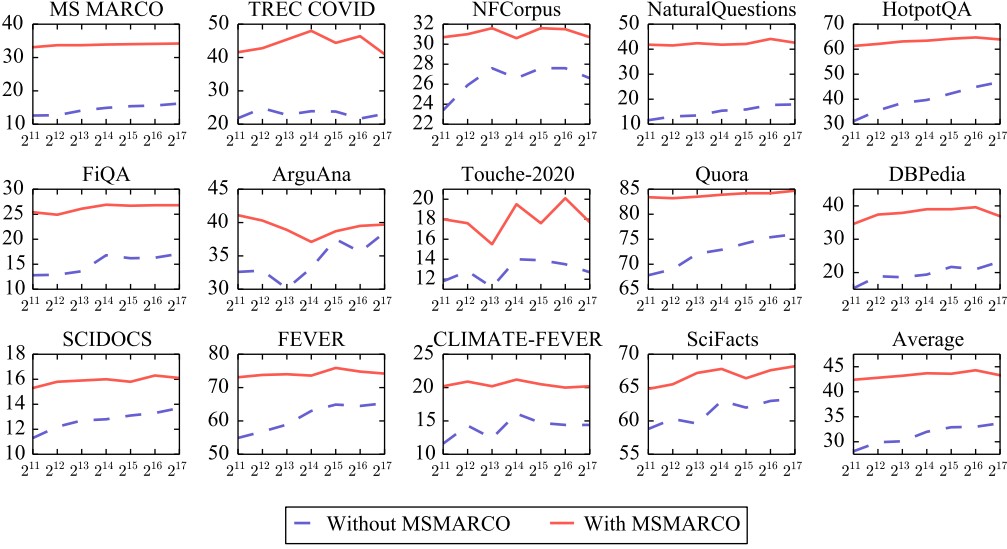

Figure 2: **Impact of the number of negatives.** We report nDCG@10 as a function of the queue size, with and without fine-tuning on MS-MARCO. We report numbers using the MoCo framework where the keys for the negatives are computed with the momentum encoder and stored in a queue.

**Data augmentations.** Third, we compare different ways to generate pairs of positive examples from a single document or chunk of text. In particular, we compare random cropping, which leads to pairs with overlap, and the inverse cloze task, which was previously considered to pre-train retrievers. Interestingly, as shown in Table 4, the random cropping strategy strongly outperforms the inverse cloze task in our setting. We argue that overlap between the two views of the data is beneficial, as it allows the network to learn lexical matching, similarly to BM25. We also investigate

whether additional data perturbations, such as random word deletion or replacement, are beneficial for retrieval. After fine-tuning on MS-MARCO, it seems that such data perturbations slightly degrade the performance of dense retrievers.

|  | NFCorpus | NQ | ArguAna | Quora | DBPedia | SciDocs | FEVER | Overall |
|---|---|---|---|---|---|---|---|---|
| ICT | 30.5 | 37.5 | 32.6 | 82.3 | 33.1 | 14.6 | 67.9 | 38.8 |
| Crop | 31.5 | 44.1 | 39.5 | 84.2 | 39.6 | 16.3 | 74.8 | 43.4 |
| Crop + delete | 31.6 | 42.3 | 39.2 | 84.6 | 39.7 | 16.2 | 71.8 | 42.6 |
| Crop + replace | 30.5 | 43.2 | 42.6 | 84.6 | 39.2 | 16.0 | 73.8 | 42.8 |

Table 4: **Impact of data augmentions.** nDCG@10 after fine-tuning on MS-MARCO.

**Training data.** Finally, we study the impact of the pre-training data on the performance of our retriever, by training on Wikipedia, CCNet or a mix of both sources of data. We report results in Table 5, and observe that there is no clear winner between the two data sources. Unsurprisingly, pre-training on Wikipedia leads to better performance on datasets for which documents also come from Wikipedia, such as NaturalQuestions or FEVER. On the other hand, on datasets from different domains than Wikipedia such as FiQA or ArguAna, training on the more diverse CCNet data leads to better results. To get the best of both worlds, we consider to strategies to mix the two data sources. In the "50/50%" strategy, examples are sampled uniformly across domain, meaning that half the batches are from Wikipedia and the other half from CCNet. In the "uniform" strategy, examples are sampled uniformly over the union of the dataset. Since CCNet is significantly larger than Wikipedia, this means that most of the batches are from CCNet.

|  | NFCorpus | NQ | FiQA | ArguAna | Quora | DBPedia | SciDocs | FEVER | Overall |
|---|---|---|---|---|---|---|---|---|---|
| Wiki | 31.5 | 44.1 | 26.8 | 39.5 | 84.2 | 39.6 | 16.3 | 74.8 | 43.4 |
| CCNet | 31.8 | 39.9 | 31.6 | 43.3 | 84.5 | 36.1 | 16.0 | 66.4 | 41.8 |
| Uniform | 32.6 | 40.6 | 31.0 | 42.3 | 84.8 | 37.2 | 15.9 | 66.5 | 42.6 |
| 50/50% | 32.5 | 41.4 | 29.4 | 40.4 | 84.6 | 39.0 | 16.3 | 75.4 | 43.9 |

Table 5: **Training data.** We report nDCG@10 after fine-tuning on MS-MARCO.

**Impact of fine-tuning on MS-MARCO.** To isolate the impact of our pre-training method as opposed to the impact of our fine-tuning on MS-MARCO, we propose to apply the same fine-tuning to the BERT base uncased model. We report results in Table 6, and observe that when applied to BERT, our fine-tuning leads to results that are lower than the state-of-the-art. Hence, we believe that most of the improvement compared to the state-of-the-art retrievers can be attributed to our contrastive pre-training strategy.

|  | NFCorpus | NQ | FiQA | ArguAna | Quora | DBPedia | SciDocs | FEVER | Overall |
|---|---|---|---|---|---|---|---|---|---|
| BERT | 28.2 | 44.6 | 25.9 | 35.0 | 84.0 | 34.4 | 13.0 | 69.8 | 42.0 |
| Ours | 33.2 | 50.2 | 28.8 | 46.0 | 85.4 | 38.8 | 16.0 | 77.7 | 46.5 |

Table 6: **Finetuning.** We report nDCG@10 after fine-tuning BERT and our model on MS-MARCO.

## DISCUSSION

In this work, we propose to explore the limits of contrastive pre-training to learn dense text retrievers. We use the MoCo technique, which allows us to train with a large number of negative examples. We make several interesting observations: first, while the inverse Cloze task has been introduced in the context of retrieval, we show that independent random cropping leads to better results. Second, we show that neural networks trained without supervision using contrastive learning exhibits good (albeit not state-of-the-art) retrieval performance. Based on these observations, we trained a model that obtains state-of-the-art performance on the BEIR benchmark when fine-tuned on MS-MARCO.

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

| | BM25 | ColBERT | SPARTA | docT5Query | DPR | ANCE | GenQ | TAS-B | Ours |
|---|---|---|---|---|---|---|---|---|---|
| TREC-COVID | 49.8 | 46.4 | 40.9 | **54.1** | 21.2 | 45.7 | 45.6 | 38.7 | 42.2 |
| NFCorpus | 25.0 | 25.4 | 24.3 | 25.3 | 20.8 | 23.2 | 28.0 | 28.0 | **28.6** |
| NQ | 76.0 | 91.2 | 78.7 | 83.2 | 88.0 | 83.6 | 86.2 | 90.3 | **92.4** |
| HotpotQA | 74.0 | 74.8 | 65.1 | 70.9 | 59.1 | 57.8 | 67.3 | 72.8 | **78.6** |
| FiQA | 53.9 | 60.3 | 44.6 | 59.8 | 34.2 | 58.1 | **61.8** | 59.3 | 59.6 |
| ArguAna | 94.2 | 91.4 | 89.3 | 97.2 | 75.1 | 93.7 | **97.8** | 94.2 | 97.7 |
| Touché-2020 | 53.8 | 43.9 | 38.1 | **55.7** | 30.1 | 45.8 | 45.1 | 43.1 | 30.8 |
| CQADupStack | 60.6 | 62.4 | 52.1 | 63.8 | 40.3 | 57.9 | **65.4** | 62.2 | 63.9 |
| Quora | 97.3 | **98.9** | 89.6 | 98.2 | 47.0 | 98.7 | 98.8 | 98.6 | **98.9** |
| DBPedia | 39.8 | 46.1 | 41.1 | 36.5 | 34.9 | 31.9 | 43.1 | 49.9 | **53.6** |
| SCIDOCS | 35.6 | 34.4 | 29.7 | 36.0 | 21.9 | 26.9 | 33.2 | 33.5 | **37.2** |
| FEVER | 93.1 | 93.4 | 84.3 | 91.6 | 84.0 | 90.0 | 92.8 | 93.7 | **95.1** |
| Climate-FEVER | 43.6 | 44.4 | 22.7 | 42.7 | 39.0 | 44.5 | 45.0 | 53.4 | **56.0** |
| SciFact | 90.8 | 87.8 | 86.3 | 91.4 | 72.7 | 81.6 | 89.3 | 89.1 | **92.2** |
| Avg. | 63.4 | 64.3 | 56.2 | 64.7 | 47.7 | 60.0 | 64.2 | 64.8 | **66.2** |
| Best on | 0 | 1 | 0 | 1 | 0 | 0 | 3 | 0 | **9** |

Table 7: **BEIR Benchmark.** We report Recall@100 on the test sets of the datasets from the BEIR benchmark for bi-encoder methods without re-ranker. We also report the average ("Avg.") and number of datasets where a method is the best ("Best on") over the entire BEIR benchmark (excluding three datasets because of their licence). We evaluate retrievers after pre-training on unsupervised data with our contrastive learning and finetuning on MS-MARCO ("Ours"). Bold is the best overall.

## A TECHNICAL DETAILS

**Contrastive pre-training.** For the model with fine-tuning on MS-MARCO (respectively, without fine-tuning), we use the MoCo algorithm He et al. (2020) with a queue of size 65536 (resp. 131,072), a momentum value of 0.999 (resp. 0.9995) and a temperature of 0.05. We use the random cropping data augmentation, with documents of 256 tokens and span sizes sampled between 10% and 50% (resp. 5% and 50%) of the document length. We also apply word deletion with a probability of 20%. We optimize the model with the AdamW (Loshchilov & Hutter, 2019) optimizer, with learning rate of $10^{-4}$ (resp. $5 \cdot 10^{-5}$), batch size of 2,048 and 500,000 (resp. 200,000) steps. We initialize the network with the publicly available BERT base uncased model. For training data, we use a data from Wikipedia (resp. a mix between Wikipedia and CC-net data (Wenzek et al., 2020), where half the batches are sampled from each source).

**Fine-tuning on MS-MARCO.** When fine-tuning our model on MS-MARCO, we use the ASAM optimizer (Kwon et al., 2021), with a learning rate of $10^{-5}$ and a batch size of 1024 with a temperature of 0.05, also used during pretraining. We train an initial model with random negative examples for 20000 steps, mine hard negatives with this first model, and re-train a second model with those. Each query is associated with a gold document and a negative document, which is a random document in the first phase and a hard negative 10% of the time in the second phase. All documents from a batch are used as negatives for each query.

| | NaturalQuestions | | | | TriviaQA | | | |
|---|---|---|---|---|---|---|---|---|
| | R@1 | R@5 | R@20 | R@100 | R@1 | R@5 | R@20 | R@100 |
| Inverse Cloze Task (Sachan et al., 2021b) | 12.6 | 32.3 | 50.9 | 66.8 | 19.2 | 40.2 | 57.5 | 73.6 |
| Masked salient spans (Sachan et al., 2021b) | 20.0 | 41.7 | 59.8 | 74.9 | 31.7 | 53.3 | 68.2 | 79.4 |
| BM25 (Ma et al., 2021) | - | - | 62.9 | 78.3 | - | - | **76.4** | **83.2** |
| Ours | **21.9** | **47.2** | **67.2** | **81.3** | **33.9** | **59.5** | 74.2 | **83.2** |
| Supervised topline: DPR (Karpukhin et al., 2020) | - | - | 78.4 | 85.4 | - | - | 79.4 | 85.0 |

Table 8: Recall@k on the test sets of NaturalQuestions and TriviaQA. For Inverse Cloze Task and Masked Salient Spans we report the results of Sachan et al. (2021b). The Masked Salient Spans model uses annotated entity recognition data.

