# OpenReview forum: "Contrastive Pre-training for Zero-Shot Information Retrieval"
_ICLR.cc/2022/Conference — ICLR 2022 Submitted_

### Official Review · Reviewer_cPwk · 2021-10-26

**Correctness:** 3
**Technical Novelty And Significance:** 2
**Empirical Novelty And Significance:** 2
**Recommendation:** 3
**Confidence:** 4

**Main Review:**

Pros:
- A cropping-based positive pair construction method is proposed and compared with the ICT-based method.
- The zero-shot evaluation and comparison on the BEIR benchmark look good.

Cons:
- Using contrastive learning in dense IR is not new. Most technical pieces mentioned here (SimCLR, MoCo, ICT, etc.) were used somewhere else before.
- For the fine-tuning setting (e.g. Table 6), it could be better if the baseline dense IR approaches (mentioned in Table 1) can be included as well, instead of just comparing with BERT.


**Summary Of The Paper:**

This paper proposed a contrastive learning-based method for zero-shot dense IR. Based on the contrastive learning framework, 3 positive pair construction methods and 2 negative pair construction methods are used and compared. Evaluations are performed on the BEIR benchmark, with the best average result achieved compared to a list of IR (both sparse and dense) baselines in the zero-shot setting.

**Summary Of The Review:**

Duo to the novelty and insufficient experiment, I think the paper should be further refined and improved.

---

> ### Author Response · Authors · 2021-11-23
> **Response to Reviewer cPwk**
>
>  We thank the reviewer for their comments.
> - We have updated our paper to better reflect prior works. We have also added results of other unsupervised methods on question answering datasets in Table 8, and in Figure 1 on the BEIR benchmark. It appears that in both settings our method outperforms other alternatives.
> - We would like to emphasize the fact that our goal is not to develop new techniques, but to determine how far we can push unsupervised pre-training for IR. In particular, we believe that a novel aspect of our work is to show that we can obtain unsupervised performance close or on par with BM25.
> - The numbers reported in Table 6 are comparable to the number reported in Table 1, where other dense retrievers are included. The goal of Table 6 is to show that our improvement comes from the contrastive pretraining and not from the fine-tuning procedure on MSMARCO.

---

### Official Review · Reviewer_F8YA · 2021-11-03

**Correctness:** 2
**Technical Novelty And Significance:** 2
**Empirical Novelty And Significance:** 2
**Recommendation:** 3
**Confidence:** 5

**Details Of Ethics Concerns:**

Introduction section and Related work sections should be expanded to cite all the previous works in the field of unsupervised text retrieval. Please see my review listed under Weakness where I also provide suggestions to the authors on how it can be done.

**Main Review:**

The proposed model has some merits but a lot of weaknesses, which I elaborate next.

**Weaknesses**:
The major weaknesses in the paper are:

Introduction Section:
1. The authors need to assign much more credit to the previous papers on unsupervised retrieval using contrastive methods. It is sad to see that the authors have not properly cited even the original ICT paper which was the first to propose the zero-shot neural text retrieval model and several important follow up works ([1], [2], [3]) in the Introduction section. The authors should pay careful attention to how they are positioning their work in relation to the previous well-known work on this topic. It’s appalling to see this style of writing that leads to improper credit assignment. I am sorry to say but writing the paper in a way where you intentionally or unintentionally hide the contributions of the previous work on the topic misguides the reader towards the contributions of the paper and is not in the spirit of a top-tier conference like ICLR.

2. From the Introduction section, the paper positions itself as a new unsupervised contrastive representation learning algorithm in the field of neural retrieval. My suggestion to you would be to position the paper as a followup to the ICT and subsequent works. And then explain clearly the new features in your proposed model when compared against a strongly trained ICT and followup models in references ([1], [2], [3]). You can make a table in the Introduction section listing the new introduced aspects in the proposed model and compare them in a rigorous manner with the previous work.

Related Work Section:

3. In the 4th paragraph, it is wrongly mentioned that Karpukhin et al. 2020 in their DPR paper introduced the dense retriever trained with BERT initialization. This is not correct. It was first done by the ICT paper and later DPR applied their model for supervised contrastive training.
Minor edits: (Izacard and Grave, 2021) and (Yang and Seo, 2020) appear two times in the same sentence. My suggestion would be to cite all the important work in unsupervised dense retrieval such as [1, 2, 3] etc. and then contrast your work with the results of previous work.

Other issues in writing

4. Section 3.1.2 first paragraph: A method from computer vision literature is presented for the two views of an image but its citation is not provided.

Methods Section

5. Section 3.1.2: ICT paragraph: There are a couple of errors in how the ICT model is described. First, it should be clearly mentioned that the the view 1 is one sentence while the view 2 is all the remaining sentences in the paragraph. Please refer to Section 4 of the ICT paper for full details (https://arxiv.org/pdf/1906.00300.pdf). Second, to enable lexical matching in ICT, the view 1 is removed with a probability of 0.9 and is kept otherwise. Again, please see Section 4 in the original paper for more details and Section 9.2 for the analysis of why this is important during ICT training.

6. Section 3.1.3: *Negative pairs within a batch*: this method for contrastive learning was first used in ICT and subsequent work such as [2, 3]. *Negative Pairs Across Batches*: This method (MuCo) has also been investigated for dense retriever training by [4]. These papers should be cited here. Also, [4] reports some issues

Results Section:

7. Section 4.3: (Second last paragraph)  The better performance of contrastively trained models when compared to BERT in both zero-shot and few-shot settings has already been investigated earlier. For example, [3] compares the zero-shot retrieval performance for both BERT and ICT in Table 3 and for few-shot performance in Figure 3. This should be mentioned here.

8. I feel that the inclusion of zero-shot results Figure 1 in the paper doesn’t really add much value until compared to a couple of previous baselines such as that of ICT. As the code is not submitted with the paper, I can't verify if the implementation of ICT is correct or not. My suggestion is that you should first make sure the implementation of ICT is correct, then train it with the same settings as your proposed approach of Independent Cropping. Also include the results of ICT in Table1, Table2, and Figure1. This will help the reader to better appreciate the results in the paper.

**Strengths**

1. The proposed method of Independent Cropping for unsupervised retrieval training seems novel and often outperforms strong BM25 baseline on the BEIR benchmark.

2. The results presented in Table 3 are important as it demonstrates that MoCo can be leveraged to train models instead of SimCLR in a memory efficient manner.

3. If the ICT implementation is correct, then the results in Table 4 would be very insightful. Then, the authors should really dig deep into the reasons why the model performance is better than ICT when keeping all the training settings the same such as the number of training steps, batch size, etc.


**Other Questions**

1. It is mentioned that you averaged the token-level representations to compute query and document representations. Did you compare how much the results differed when you just took their [CLS] token representations?

2. Could you also report the zero-shot Recall@20,100 results in Table 4 without finetuning of MS-MARCO?

3. For qualitative analysis, can the authors present some examples on NQ when their model shows better top-1 recall than ICT due to a better lexical match?

References:

[1] Wei-Cheng Chang, Felix X. Yu, Yin-Wen Chang, Yim- ing Yang, and Sanjiv Kumar. 2020. Pre-training tasks for embedding-based large-scale retrieval. In International Conference on Learning Representations.

[2] Luan, Yi, Jacob Eisenstein, Kristina Toutanova, and Michael Collins. 2021. "Sparse, Dense, and Attentional Representations for Text Retrieval." Transactions of the Association for Computational Linguistics.

[3] Sachan, Devendra Singh, Mostofa Patwary, Mohammad Shoeybi, Neel Kant, Wei Ping, William L. Hamilton, and Bryan Catanzaro. 2021. "End-to-end training of neural retrievers for open-domain question answering." In ACL 2021.

[4] Yang, Nan, Furu Wei, Binxing Jiao, Daxing Jiang, and Linjun Yang. 2021. "xMoCo: Cross momentum contrastive learning for open-domain question answering." In ACL 2021.


**Summary Of The Paper:**

This paper presents a contrastive learning approach for unsupervised text retrieval. As the approach is unsupervised, similar to the Inverse Cloze Task (ICT) approach, the authors propose to do independent cropping of a paragraph to generate a pair of "pseudo-question" and "pseudo-document", which is used to train the model. The model is trained using contrastive learning following the MoCo algorithm. Experiments are performed on the recent BEIR benchmark for zero-shot and few-shot retrieval evaluation. The proposed approach often performs better than the BM25 algorithm on zero-shot retrieval on the BEIR benchmark. There are also some ablations presented to analyze different aspects of model training such as the effect of number of negative examples, training data, data augmentations etc.


**Summary Of The Review:**

The paper proposes the approach of independent cropping of a paragraph into two segments for unsupervised training of the dense retriever. The authors argue that independent cropping enables the model to possess lexical matching ability, a property which improves retrieval performance.

First, I feel that the writing in the Introduction and Related Work sections needs major improvements, specially that of crediting and citing the previous work in the field of unsupervised contrastive training. I have also provided my suggestions for the same under Weakness section. Second, I feel that the authors should correctly implement a ICT following the original paper and perform training using the same settings as done for their model. Third, instead of comparing with BM25, the emphasis should be on comparing results with ICT and providing an in-depth understanding of why or why not their proposed model works better.

Due to these shortcomings, in its current form, the paper falls substantially short of the ICLR publication bar.
However, I would request the authors to provide their responses to the points raised in the Weakness section and under the Other Questions.

---

> ### Author Response · Authors · 2021-11-23
> **Response to reviewer F8YA**
>
> First, we would like to thank the reviewer for the detailed review and feedback. As mentioned in the general response to the reviewers, we have updated the introduction and related work sections to better position our paper with respect to prior work. It was not our intention to hide contributions of previous work on the topic, and we made that more clear in the paper.
>
> Regarding results:
> - We have added comparisons to REALM and SimCSE on the BEIR benchmark, using the models available on HuggingFace. For SimCSE, we report results with RoBERTa-Large (trained without supervised data), while REALM was trained using named entity recognition data (for the saliency masking loss function). Our model compares favorably to these two unsupervised models.
> - On NaturalQuestions and TriviaQA, two competitive question answering datasets, we report results of unsupervised dense retrievers from [1], including ICT and saliency masking, and compare our model to these. Again, our model compares favorably, outperforming these previous results while being competitive with BM25.
>
> Regarding the reproduction of ICT, we thank the reviewer for highlighting the discrepancy between the way this task is described in our paper, and in the original paper. We fixed the description and implementation, but early experiments seem to indicate that the conclusion of our paper still holds:
> - When used with the MoCo method, the training does not converge well, probably due to the fact that the task is asymmetric (queries and keys do not have the same distribution), which is problematic because only the query encoder is trained with backpropagation. This leads to poor unsupervised retrieval performance.
> - When using SimCLR, the results are improved a bit, but still lags behind the best setting we can obtain with MoCo and random cropping.
> - It seems that sometimes having a “gap” between the query and the surrounding context also leads to better performance, which is a property of random cropping, but not the original ICT task.
> - Finally, as indicated in the previous “results” section of this rebuttal, when comparing to existing implementations of ICT from previous work, our model obtain stronger performance (although not being an “apple-to-apple” comparison, we still believe this is additional evidence that our training procedure leads to strong unsupervised retrieval).
>
> We hope that the improved writing and the added comparisons to previous work, including ICT, show the merits of our model for unsupervised dense retrieval.
>
> [1] Devendra Singh Sachan, Siva Reddy, William Hamilton, Chris Dyer, and Dani Yogatama. End-to-end training of multi-document reader and retriever for open-domain question answering, 2021b.
>
> [2] Xueguang Ma, Kai Sun, Ronak Pradeep, and Jimmy Lin. A replication study of dense passage retriever, 2021.

---

> > ### Comment · Reviewer_F8YA · 2021-11-30
> > **Review Update**
> >
> > Thanks to the authors for updating the paper and for adding some new results.
> >
> > I still feel that the paper writing can be improved a lot including the Introduction and Related Work sections. For example, Table 8 needs to be in the main paper and not in the appendix. There can be a better comparison with the previous methods to highlight the novelty in the paper. In my opinion, the focus should be on answering "why the proposed technique works better than ICT or REALM" rather than doing an exhaustive comparison with BM25. The authors can also highlight what can be the takeaways of this work for future research in this area.
> >
> > Due to these, I am keeping my scores unchanged but urge the authors to extensively revise the writing of the paper including a fairer comparison with previous work and then resubmit.

---

### Official Review · Reviewer_NWVD · 2021-11-07

**Correctness:** 3
**Technical Novelty And Significance:** 2
**Empirical Novelty And Significance:** 2
**Recommendation:** 5
**Confidence:** 4

**Main Review:**

The paper addresses an important problem for neural models for IR: what unsupervised training task is suitable for an IR model? The existing strategies such as masked language model and ICT fit some of the NLP tasks, but not necessarily the IR task. This paper proposes to use cropping to train a model for IR.
The proposed strategy is interesting and intuitively more relevant to IR tasks. The experiments show that this is indeed better than the other training strategies.
The paper is well written, and contains a set of experiments on several datasets. The experimental results are convincing.
One broader question is on the idea of replacing BM25 in the zero-shot learning scenario. As the experiments show, when no training data (aligned query-documents) is available, the proposed method can just catch up with BM25. In other words, the cropping training strategy can just reach the level of BM25, but with heavy pre-training. It does not seem reasonable to try to compete with an inexpensive method such as BM25 by a heavy model. Instead, the neural model should seek to complement BM25, i.e. capture other matching signals.
From this perspective, trying to beat BM25 in a zero-shot learning scenario may not be useful in practice.

**Summary Of The Paper:**

This paper proposes to use cropping strategy to train a neural model for IR. It is compared to BM25 and several other neural models, including BERT and the ones using inverse cloze task. The goal of cropping is to train a model that fits better the IR tasks. It is hypothesized that this task is more similar to IR than other pre-training strategies.
The method is tested on a set of benchmarks of zero-shot retrieval and MSMARCO with training data. It is shown than the proposed method fine-tuned on MSMARCO can outperform the compared methods, including BM25. When no fine-tuning is used, the method is almost at par with BM25.


**Summary Of The Review:**

The proposed method is interesting and the experiments show it effectiveness in the scenario of zero-shot learning and few-shot learning.
However, the question is about the necessity to build a complexe neural model to compete with much more efficient BM25 model in the zero-shot learning situation.

---

> ### Author Response · Authors · 2021-11-23
> **Response to Reviewer NWVD**
>
> We thank the reviewer for their comments.
>
> While we agree that currently, purely unsupervised dense retrievers may not be widely applicable in practice compared to BM25, we also believe that this is a promising research direction, and that dense retrievers will likely become more common for zero shot learning scenarios in the future. Indeed, for some applications such as question answering, unsupervised dense retrievers are already comparing favorably to BM25 (eg. see Table 8 of our updated paper, where our model obtains +3.0 R@100 on NaturalQuestions). Additionally, when retrievers are used for such downstream tasks (eg. QA), unsupervised dense models can be improved, for example by using distillation from the reader model, as opposed to BM25. This is also the case when few examples are available (few-shot setting), where the zero-shot performance seems to correlate well with the few-shot performance.

---

### Official Review · Reviewer_tnYn · 2021-11-08

**Correctness:** 3
**Technical Novelty And Significance:** 2
**Empirical Novelty And Significance:** 2
**Recommendation:** 5
**Confidence:** 3

**Main Review:**

## Strengths
- the paper writing is clear and easy to follow
- solid empircal comparison

## Weaknesses
- lack of techincal novelty. The contrastive loss for learning Siamese Transformers is not new. Creating pseudo (query, doc) pairs for the constrastive loss is also studied in ICT.
- the ablation study of truely zero-shot (w/o fine-tuning with MS-MARCO) evaluation on BEIR is missing.
- the author should also compare with other pre-training frameworks in Figure 1, including ICT, REALM, and SimCSE.
- from Figure 1, it seems like the the proposed model is worse than BM25 in terms of the average Recall@100
- for reproducibility, the author should release the experiment code, pre-training datasets and the pre-trained models




**Summary Of The Paper:**

This paper studies pre-training of Siamese Transformer encoders for the zero-shot dense retrieval problem. The author consider contrastive learning loss function to optimize the encoders and investigate variants of ICT for data augmentations. While not much novelty in the techincal sides, the author indeed present solid empirical study on zero-shot retrieval benchmark as well as extension to the few-shot evaluation.

**Summary Of The Review:**

This paper studies pre-training of Siamese Transformer encoders for the zero-shot dense retrieval problem. While the empirical results seem promising, there are very little techincal novelty in this work. Since this paper concerns pre-training tasks for retrieval, the author should also show the results of ICT and REALM for the Figure 1 setup. At the moment, I think this paper is marginally below the acceptance threshold. If the author can address my concern about proper ablation comparisons, I am very happly to update the recommendation scores.

---

> ### Author Response · Authors · 2021-11-23
> **Response to Reviewer tnYn**
>
> We thank the reviewer for their comments.
> - We’ve updated our paper to better reflect prior works using contrastive learning for training dense retrievers.
> - We’ve added the performance of REALM and SimCSE on the BEIR benchmark in Figure 1, and results on question answering datasets in Table 8 following the setting used in [1]. Our model compares favorably against previous works in both settings. Interestingly on question answering datasets, our model is on par with or better than the strong BM25 model obtained in [2].
> - It’s true that the average Recall@100 is worse for our unsupervised model than for BM25, but as noted in the text this comes from the large difference between BM25 and our method on two datasets: webis-touche2020 and TREC-Covid (this is also true for other dense retrievers, including supervised ones). Please note that our unsupervised retriever outperforms BM25 for the Recall@100 on 8 out of 15 of the datasets. Finally, the sensitivity is different from one dataset to another, thus the average across datasets is not ideal.
> - We will add the same ablation study for zero-shot performance and we will release our code and pretrained models.
>
> [1] Devendra Singh Sachan, Siva Reddy, William Hamilton, Chris Dyer, and Dani Yogatama. End-to-end training of multi-document reader and retriever for open-domain question answering, 2021b.
>
> [2] Xueguang Ma, Kai Sun, Ronak Pradeep, and Jimmy Lin. A replication study of dense passage retriever, 2021.

---

### Author Response · Authors · 2021-11-23
**General response to the reviewers**

We would like to thank all reviewers for their helpful feedback. Here is a short summary of the main changes:
- We’ve updated the introduction and the related work to better position our paper in the context of prior work.
- We have added results on question answering datasets in Table 8. Our model matches BM25 performance, and outperforms other unsupervised dense retrievers.
- We have added the performance of SimCSE and REALM on the BEIR benchmark in Figure 1. Similarly to question answering datasets, our model outperforms previous unsupervised models.

---

### Decision · Program_Chairs · 2022-01-20

**Decision:**

Reject

**Comment:**

Good premise:  What unsupervised training supports IR?  This is a key question for IR and is a focus for papers in TREC 2019 Deep Learning Track, for instance.  Also, historically, empirical work in the IR community is a very high standard.

One reviewer says the contrastive loss for learning Siamese Transformers is not new, and prior experimental work was listed.  Several reviewers suggested extensions to the empirical work, some of which was subsequently done.  Results are "promising" according to one reviewer, but not strong enough.  Another reviewer says a different use context is needed since its hard to compete with efficient BM2t in its own terms.  The authors made some good changes to their paper: updated intro and related work, extended results and discussion, but the 4 reviewers remained in agreement, a reject.  However, some reviewers felt this was a good contribution, so with further empirical work and polish it should be good.